# Influence of Clones on Relationship between Natural Rubber and Size of Rubber Particles in Latex

**DOI:** 10.3390/ijms23168880

**Published:** 2022-08-10

**Authors:** Simin He, Fuquan Zhang, Fengshou Gu, Tianqi Zhao, Yanfang Zhao, Lusheng Liao, Xiaoxue Liao

**Affiliations:** 1Key Laboratory of Tropical Crop Products Processing of Ministry of Agriculture and Rural Affairs, Agricultural Products Processing Research Institute of Chinese Academy of Tropical Agricultural Sciences, Zhanjiang 524001, China; 2School of Materials Science and Engineering, Hainan University, Haikou 570228, China; 3Hainan Provincial Key Laboratory of Natural Rubber Processing, Zhanjiang 524001, China

**Keywords:** clone, rubber particle size, molecular weight, nonrubber component content

## Abstract

IAN873, Dongfang93114 and Reyan73397, created through vegetative propagation for their high yield and excellent cold resistance, are major clones planted in China. In this work, latexes with rubber particles of the same size from these clones are separated from fresh natural rubber latex, and corresponding rubber films are prepared from each latex. The structure and components of each film are measured. This indicates that the characteristics of the rubbers obtained from latexes with similar particle sizes show some resembling trends among different clones, while for specific samples, those characteristics vary depending on the clone. The molecular weight is generally highest in IAN873 and lowest in Reyan73397. Rubber chains in small rubber particles are longer, and large rubber particles show a wider molecular weight distribution. The gel content of every sample from Reyan73397 is lower than the other two clones. The nitrogen content increases with the size of rubber particles in all clones. The ester content of small rubber particles in IAN873 and Reyan73397 is almost zero. Large rubber particles have more branching points formed via esters. This study provides a new perspective on the influence of clones on the relationship between characteristics of natural rubber and the size of rubber particles in natural rubber latex.

## 1. Introduction

Natural rubber (NR), an elastic solid composed of natural rubber latex (NRL) through solidification, drying and other processing procedures, is of great use due to its comprehensively excellent properties that are superior to synthetic rubber [1,2,3,4]. For example, NR plays an irreplaceable role in the tread rubber of airplane tires [5,6,7]. The *Hevea brasiliensis* rubber tree is currently almost the exclusive commercial source of NR [8,9,10]. As a biosynthetic polymer, the structures and properties of NR are affected by a great number of biological conditions, such as clones, soil, season, climate, etc. Among these factors, clones play an essential role [11,12,13,14].

Previous studies have reported that the structures and properties of natural rubber indicate obvious differences in different clones of the *Hevea brasiliensis* rubber tree. Bonfils et al. [15] characterized the mesostructure of four *Hevea brasiliensis* clones (RRIM600, GT1, RRIC110 and PB217). It was found that rubber prepared from PB217 contained heavier and more compact microaggregates than other clones. Wisunthorn et al. [16] studied the dynamic structuring of RRIM600 and PB235. They detected that the macromolecular structure, especially the Mn and gel content, dramatically increased after the fast structuring for both clones, but the explanations were different. The increase in M_n_ in RRIM600 was mainly caused by a decrease in the short chain’s number, while that of PB235 was mainly due to an increase in the microaggregates’ quantity or size. The increase in gel content resulted from macroaggregates in RRIM600 and was mostly due to microaggregates in PB235. Liengprayoon et al. [17] reported that the lipid quantity and composition were dependent on the clone. Patrini et al. [14] reported that the technological properties of raw NR, including Wallace plasticity, the plasticity retention index and Mooney viscosity, showed obvious variation between clones. PR255 presented the highest sensitivity to thermo-oxidation among GT1, PR255, FX3864 and RRIM600. The different performances of NR from various clones can meet different corresponding industrial uses.

The structures, components and properties of NR are related to the size of the rubber particles in NRL. NRL exists as a heterogeneous system, where rubber particles (RPs) are suspended in an aqueous phase containing nonrubber components (NRCs), such as proteins, esters, etc. Although NRCs account for less than 10% of NR, they play an indispensable role in the excellent characteristics of NR [18,19,20]. RPs are assembly formed with a core–shell structure. Polyisoprene exists as a core, surrounding a mixed protein−lipid layer or a lipid monolayer as a shell [21]. In addition, previous works have reported that protein can promote vulcanization and ageing resistance of natural rubber [22]. The size of rubber particles ranges from tens of nanometers to several microns, and the structures and properties vary with the particle size. Tarachiwin et al. [23] separated FNRL into several fractions with different rubber particles and studied the structures as well as components of each fraction. They found that the molecular weight, number of branching points, nitrogen (N) content and ester content varied with rubber particles. Qu et al. [24] found that small rubber particles (SRPs) with low branching numbers and branching frequency were believed to be composed of almost linear rubber molecules having no chain end groups to be branched. In contrast, LRPs possessed a high branching number, and the branch points were mainly formed through the association of phospholipids via hydrogen bonding and ionic linkages. Sriring et al. [22,25,26,27] reported that the film formation process, green properties and mechanical properties of vulcanizates varied between SRPs and LRPs.

However, rubber particles of the same size but from different clones, whether the structures, compositions and properties are the same or not, is a subject worth studying. In this work, the FNRL of IAN873, Dongfang93114 and Reyan73397 is separated into eight fractions with almost identical rubber particle sizes, and rubber films are prepared from each latex. The molecular weight, molecular weight distribution, gel content, nitrogen content and ester content of the rubber films are analyzed. Additionally, the presumed structure of representative rubber particles is portrayed. This study reveals the influence of clones on the relationship between characteristics of NR and the size of rubber particles in NRL. It provides a new perspective for adjusting the nitrogen content, ester content, etc., by selecting clones and particle sizes so that applications in different fields can be met.

## 2. Results and Discussion

### 2.1. Average Particle Size and Particle Size Distribution

The average particle size and particle size distribution of FNRL and separated latexes from the three clones are shown in Figure 1 and Table 1. It can be seen that all the FNRL samples had bimodal distributions, while the separated latexes showed unimodal distributions. Additionally, the fitting curves of the FNRL (fitted from the eight separated curves) were also bimodal. This indicated that the fractions with narrow size distributions were obtained through the centrifuging process. Through the centrifuging method, the FNRL of all clones was separated into eight grades with gradually increasing particle sizes. Latex-#1, with the smallest particle size, had an average particle size of 130–140 nm. Latex-#2 had an average particle size of 170–200 nm. These two samples, with an average particle size lower than 200 nm, were classified as SRPs. Other samples with an average particle size higher than 200 nm were classified as LRPs [28,29,30]. The average particle size of latex-#3 was in the 200-250 nm range, and that of latex-#4 was between 300 nm and 400 nm. The average particle size of latex-#5 was between 550 nm and 650 nm and that of latex-#6 was between 800 and 900 nm. Latex-#7 had an average particle between 900 and 1000 nm. Latex-#8, with the largest particle size, had an average particle size above 1 μm.

In general, not much difference was shown in the particle size distribution ranges of the FNRL from different clones. Additionally, the average particle sizes of the separated grades were also similar. However, the content of each grade varied in different clones. This indicated that the FNRL from all clones contained rubber particles in the same size range, but the size composition of rubber particles was related to the clone.

### 2.2. Apparent Color

Figure 2 shows the photos of rubber films obtained from latexes with rubber particles of various sizes. The color of the rubber films obtained from SRPs was deeper than that from LRPs. This may be attributed to the higher NRC content of SRPs than LRPs, such as protein and ester [31].

### 2.3. Molecular Weight and Gel Content

Figure 3 shows the molecular weight distribution (MWD) of rubber films obtained from separated latexes from different *Hevea brasiliensis* clones. The calculated number average molecular weight (${\overset{–\;–}{M}}_{n}$), weight average molecular weight (${\overset{–\;–}{M}}_{w}$) and polydispersity index (PDI) are shown in Figure 4. For most of the rubber films, the ${\overset{–\;–}{M}}_{w}$ of samples from Reyan73397 was the highest, and that of IAN873 was the lowest. For all clones, the ${\overset{–\;–}{M}}_{w}$ and ${\overset{–\;–}{M}}_{n}$ of SRPs were generally higher than those of LRPs. The PDI of samples obtained from LRPs was higher than those from SRPs. This indicated that SRP samples had longer rubber chains, while LRP samples showed wider MWDs. For SRPs, the molecular weight increased with the rise in particle size. The samples 2# in all clones showed the highest molecular weight. For sample 8#, with a particle size over 1 μm, the molecular weight showed an obvious increase. However, samples obtained from latex with rubber particles of similar sizes showed distinguished values of ${\overset{–\;–}{M}}_{n}$, ${\overset{–\;–}{M}}_{w}$ and PDI in different clones. In brief, some similar trends were shown, such as the length of rubber chains, and the MWD of NR changed with the rubber particle sizes among different clones. However, for specific samples, the relationship between molecular weight and the size of rubber particles varied with clone.

The gel content of rubber films obtained from separated latexes of IAN873, Dongfang93114 and Reyan73397 is shown in Figure 5. Reyan73397 showed the lowest gel content in all samples. The gel content increased with the increase in the rubber particle size in Dongfang93114, showing the most regular correlation among the three clones. Additionally, the mathematical model was:*GC* = 0.01588*s* + 33.73 (*R*^2^ = 0.9918)(1)
where *GC*: gel content; *s*: particle size.

Hence, for Dongfang93114, the gel content was adjusted by selecting rubber particles with corresponding sizes.

The gel content of IAN873 firstly increased with the increase in particle size and reached the maximum value of 174 nm (34.20%). Then, it decreased to the lowest value of 223 nm (27.88%) and, subsequently, increased to the maximum value of 619 nm (gel content 51.69%). When the average particle size was 841 nm, the gel content reached the minimum value (46.19%) and then increased slightly to 48.2% and 48.5%. The gel content of sample 3# was the lowest, and that of sample 5# was the highest.

The gel content of Reyan73397 firstly increased to the maximum value (19.77%) when the average particle size was 194 nm, and then gradually decreased to the minimum value (3.34%) when the average particle size was 326 nm. After that, it increased continuously with the increase in rubber particle size. The gel content of sample 4# was the lowest, and that of sample 8# was the highest.

These results indicated that the gel content of rubber particles of the same size varied with clone. Additionally, the relationship between the gel content and rubber particle size also varied with clones. As the gel phase was generated by links/interactions between the NRCs and polyisoprene macromolecules, the discrepancy in gel content may be related to the difference in NRCs.

### 2.4. Nonrubber Component Content

Because of the important role of NRCs in NR, the content of the main NRCs, protein and ester was tested as follows.

#### 2.4.1. Nitrogen Content

Figure 6 shows the nitrogen (N) content of the rubber films obtained from the separated latexes. For samples 1# to 8#, the N content decreased with the increase in rubber particle size. Additionally, the mathematical models were:(2)$${\mathrm{NC}}_{\mathrm{IAN}873}=\frac{1112.35}{s^{1.49}}\;(R^{2}=0.97150)$$
(3)$${\mathrm{NC}}_{\mathrm{Dongfang}93114}=\frac{543.07}{s^{1.34}}\;(R^{2}=0.9808)$$
(4)$${\mathrm{NC}}_{\mathrm{Reyan}73397}=\frac{1445.05}{s^{1.59}}\;(R^{2}=0.9573)$$
where *NC*: nitrogen content; *s*: particle size.

This was possible because rubber particles with smaller particle sizes have a larger specific surface area. Therefore, the proportion of protein in the surface film in the whole sample was larger. In summary, the relationship between the *NC* and particle size was the same in different clones, while the *NC* of rubber particles of the same size varied with clones. Therefore, the *NC* could be adjusted by selecting clones or particle sizes.

#### 2.4.2. Ester Content

Figure 7 shows the FTIR spectra of the rubber films. Additionally, the calculated ester content is listed in Figure 8. The ester content of LRPs was higher than that of SRPs in all clones. In addition, SRPs had almost no esters in IAN873 or Reyan73397. For Dongfang93114, the relationship between ester content (*EC*) and particle size could be fitted as:*EC* = 0.01888*s* + 32.63 (*R*^2^ = 0.8592)(5)
where *EC*: ester content; *s*: particle size.

For samples 1# to 8#, the ester content of samples obtained from latexes of the same rubber particle size varied with clone. This indicated that the relationship between the ester content of NR and rubber particle size in NRL was affected by the clones.

### 2.5. Presumed Structure

According to previous studies, the branching of natural rubber is mainly related to esters in the ω-terminal. Additionally, samples with few esters may have active chain ends in the ω-terminal [22,23,32]. Hence, the samples 2# for each clone were selected as representing SRPs. The samples 3# represented samples with the longest rubber chains in LRPs. Sample 4# for IAN873, sample 7# for Dongfang93114 and sample 8# for Reyan73397 represented rubbers with the most branching formed in the ω-terminal. Figure 9 shows the schematic diagram of these typical rubber samples.

As Figure 9 shows, SRPs in IAN873 and Reyan73397 had few esters; thus, there existed a lot of active chain ends, while SRPs in Dongfang93114 had a few esters; thus, there existed a number of branching points formed via esters. Additionally, the N content in SRPs was the highest. Sample 3# had the longest rubber chains with a certain content of N and esters. The ω-terminals in rubber particles with the highest ester content were almost all branched by esters.

## 3. Materials and Methods

### 3.1. Materials

Fresh natural rubber latex (FNRL) of the three clones (IAN873, Dongfang93114 and Reyan73397) were kindly provided by Guangdong Guangken Rubber Group Co., Ltd. (Guangzhou, China) Other ingredients were commercial industrial products and were used as received.

### 3.2. Sample Preparation

The FNRL of each clone was centrifuged at different speed rates into 8 fractions. The separating process is shown in Figure 10. The centrifugation process was carried out at 4 °C. All rubber latexes to be centrifuged were preadjusted to 20% dry rubber content (DRC) using a sodium dodecyl sulfate (SDS) aqueous solution at the concentration of 0.3% *w*/*w*. The separated latexes were named latex-#1 to latex-#8, respectively.

Each rubber latex was cast onto glass plates and then dried using a vacuum oven at 60 °C to obtain corresponding rubber films. The rubber films obtained from the corresponding latexes were named samples 1# to 8#, respectively.

### 3.3. Characterization

Average particle size and particle size distribution of rubber latexes were measured by using a particle size analyzer (Malvern ZSU5800, Malvern, UK) with 0.01% *v*/*v* concentration at 25 °C.

The Gel content of the rubber films was tested according to ISO/FDIS 17278:2020 (E). The test sample, cut into approximately 1 mm^3^ sized pieces, was weighed to the nearest 0.1 mg (*m*_0_). Subsequently, the pieces were placed into a clean centrifuge tube and 30 mL tetrahydrofuran (THF) was added to the bottle. After being left for 24 h in dark conditions without stirring at (25 ± 2) °C, the tube was centrifuged at 8000 rpm for 6 h. Then, the liquid and the precipitate were separated. The liquid was retained for molecular weight measurement and the precipitate was moved into a container that had been cleaned and weighed (*m*_1_). After that, the container containing the precipitate was dried at 110 °C to constant weight (the mass of the container containing the dry precipitate was denoted as *m*_2_). The gel content (*GC*) was calculated as follows:(6)$$\mathrm{GC}=\frac{\left(m_{2}{-m}_{1}\right)}{m_{0}}\;\times \;100\%$$

Weight average molecular weight (${\overset{–\;–}{M}}_{w}$), number average molecular weight (${\overset{–\;–}{M}}_{n}$), polydispersity index (PDI) and molecular weight distribution (MWD) were measured with gel permeation chromatography (GPC, Aglient 1260 Infinity, Santa Clara, CA, USA) using tetrahydrofuran (THF) as the mobile phase and polystyrene as the standard calibration at 40 °C.

The nitrogen content of the rubber films was tested by applying a Kjeldahl Analyzer (Yihon NKD-6260, Shanghai, China). The ester content of the rubber films was analyzed quantitatively using Fourier transform infrared (FTIR) spectroscopy (Bruker Tensor27, Billerica, MA, USA) [24,33]. A calibration curve was constructed by increasing the ratios of absorbances at 1739 cm^−1^ and 1664 cm^−1^ of different mixtures of methyl stearate and synthetic cis-1,4-polyisoprene, and the ester content was obtained by substituting the intensity ratio of absorbances of carbonyl groups (C=O) at 1739 cm^−1^ and unsaturated carbon (C=C) at 1664 cm^−1^ into the calibration curve equation.

## 4. Conclusions

In conclusion, this study revealed the influence of clones on the relationship between the molecular structure and the composition of NR, and the size of rubber particles in NRL. The results showed that the molecular weights were higher in LRPs than in SRPs for all clones. Additionally, the LRPs exhibited a broader MWD than SRPs. For the genotypes studied, the molecular weight of soluble rubber chains was the lowest in IAN873 and the highest in Reyan73397. The gel content of every sample from Reyan73397 was lower than its counterpart from the other two clones, while the gel content of rubber particles of the same size varied with clones. The nitrogen content increased with the increase in the size of rubber particles in all clones, and SRPs in IAN873 and Reyan73397 contained almost no esters. LRPs generally had more esters than SRPs in all clones. Sample 4# in IAN873, sample 7# in Dongfang93114 and sample 8# in Reyan73397 contained the most esters, respectively. Overall, the NRC content of samples with similar rubber particle sizes was different in every clone. To sum up, the structure and component of rubbers obtained from latexes with similar particle sizes showed some resembling trends among different clones. However, for specific samples with identical rubber particle sizes, those characteristics depended on the clone. The clones obviously influenced the relationship between the characteristics of NR and the particle sizes in NRL. This revealed a method to adjust the structure (e.g., molecular weight, molecular weight distribution and branching structure) and nonrubber component content (e.g., nitrogen content and ester content) by selecting the clone and particle size. This provides new perspectives to adjust the properties of natural rubber (e.g., cure characteristics, mechanical properties and age-resistant performance) so as to promote the application of natural rubber in different conditions through an artificial selection of natural factors (clone and particle size).

## Figures and Tables

**Figure 1 ijms-23-08880-f001:**
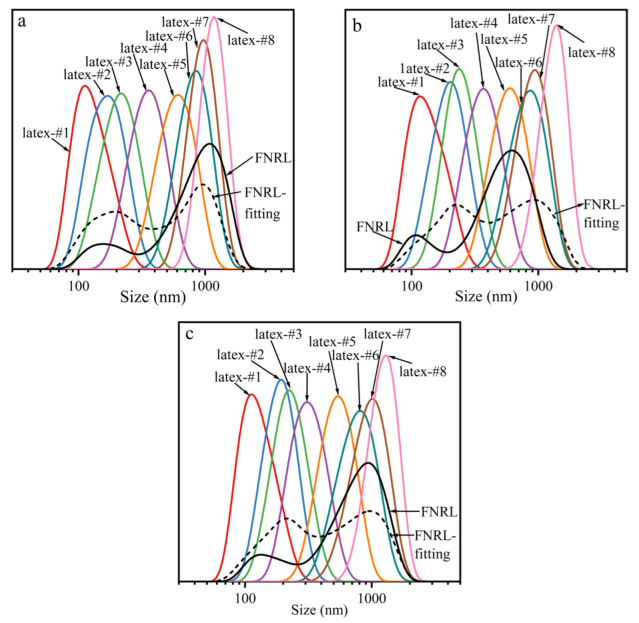
Particle size distribution of FNRL and separated latexes from different *Hevea brasiliensis* clones (arrows indicate the average particle size of the corresponding particle size distribution curve): (**a**) IAN873; (**b**) Dongfang93114; (**c**) Reyan73397.

**Figure 2 ijms-23-08880-f002:**
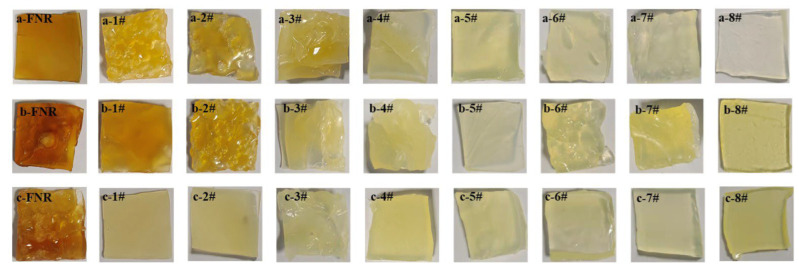
Photos of rubber films obtained from latexes with rubber particles of various sizes ((**a**) IAN873; (**b**) Dongfang93114; (**c**) Reyan73397; 1# to 8#: Rubber films obtained from latex-#1 to latex-#8).

**Figure 3 ijms-23-08880-f003:**
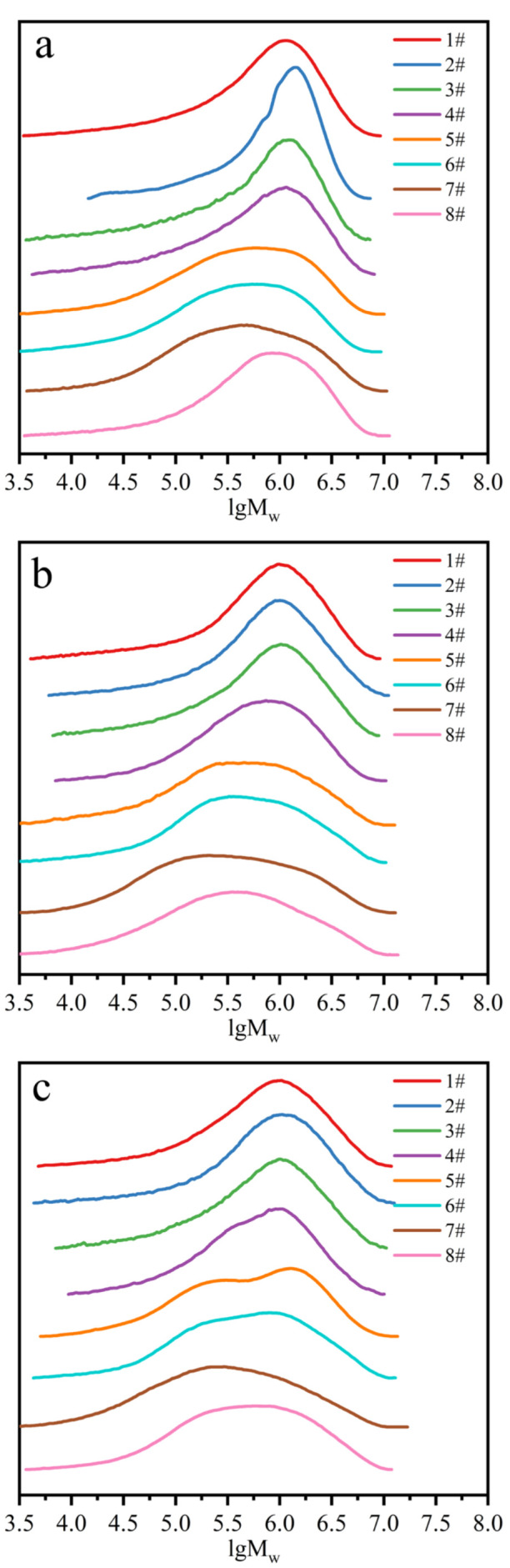
Molecular weight distributions of rubber films obtained from separated latexes from different *Hevea brasiliensis* clones: (**a**) IAN873; (**b**) Dongfang93114; (**c**) Reyan73397.

**Figure 4 ijms-23-08880-f004:**
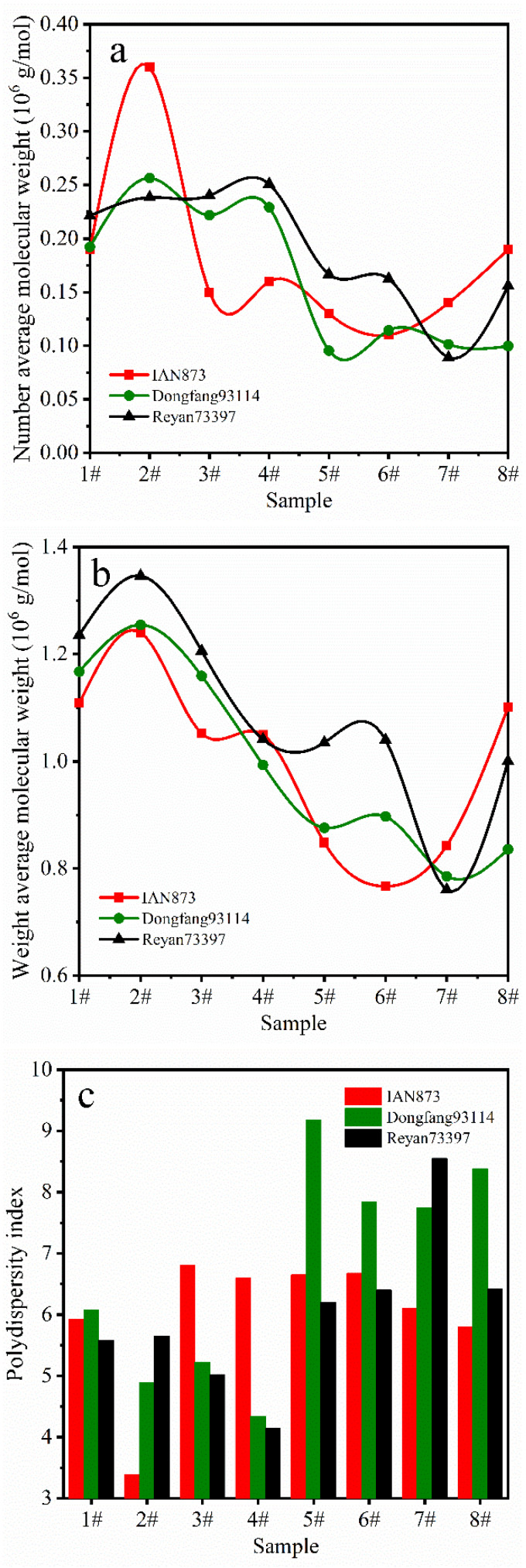
Relationships between (**a**) ${\overset{–\;–}{M}}_{n}$, (**b**) ${\overset{–\;–}{M}}_{w}$, (**c**) PDI and particle sizes of rubbers.

**Figure 5 ijms-23-08880-f005:**
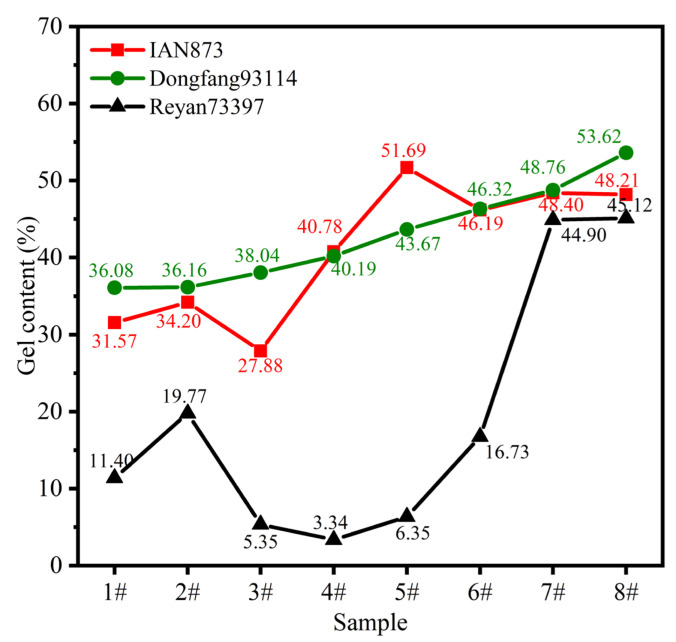
Gel content of rubber films obtained from separated latexes of different *Hevea brasiliensis* clones.

**Figure 6 ijms-23-08880-f006:**
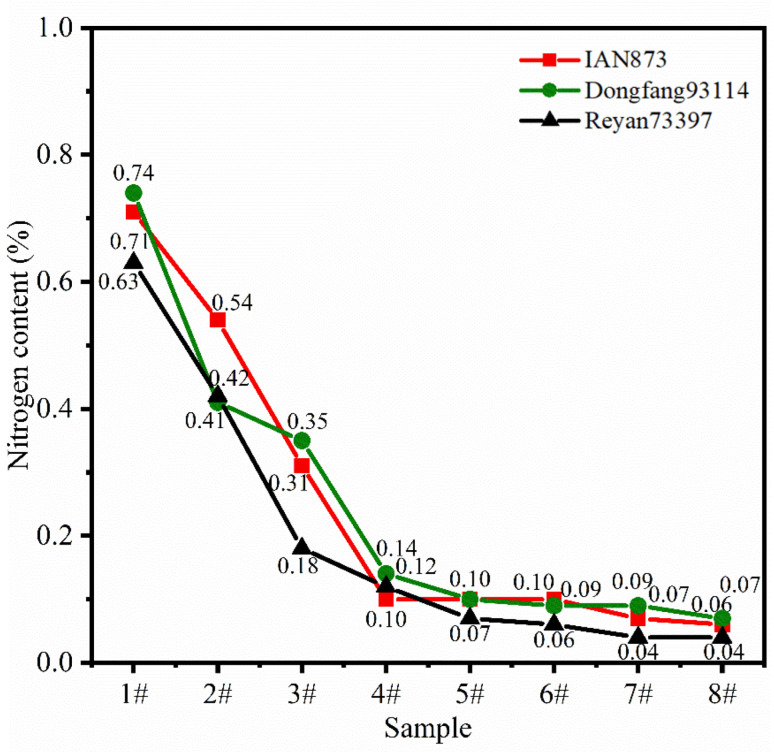
Nitrogen content of rubber films obtained from separated latexes of different *Hevea brasiliensis* clones.

**Figure 7 ijms-23-08880-f007:**
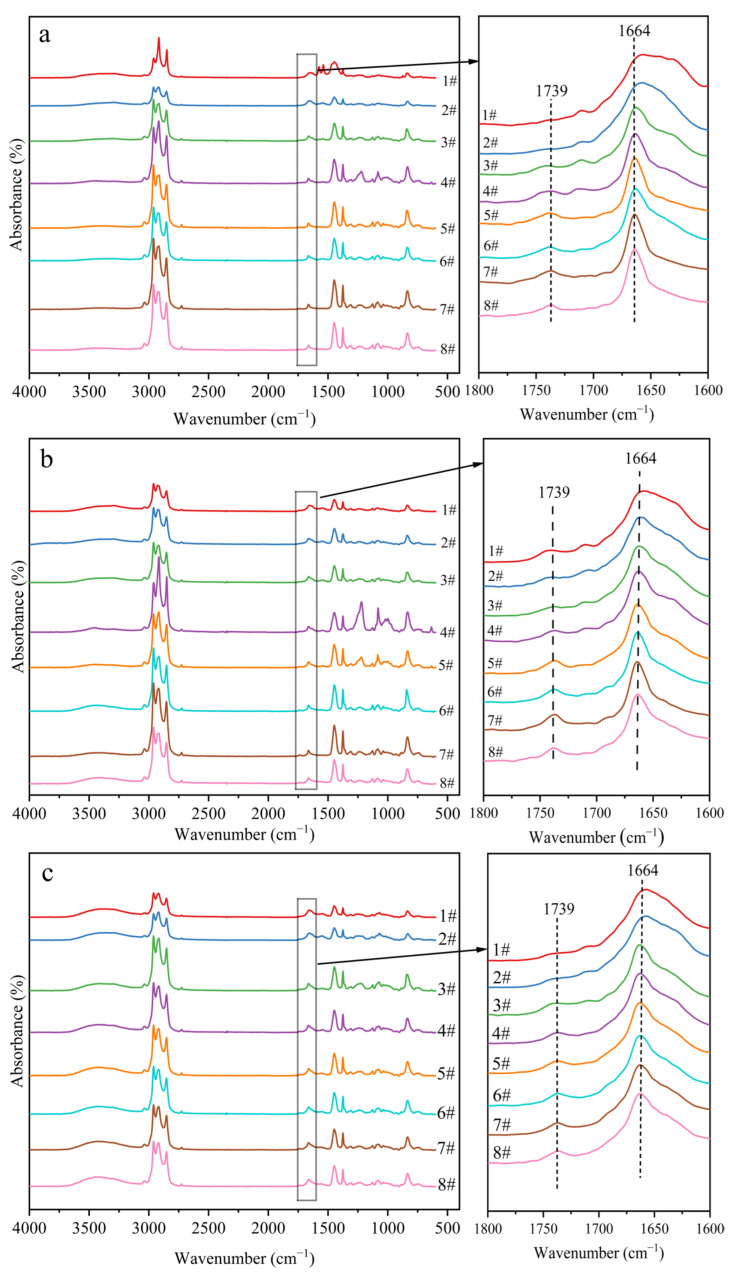
FTIR spectra of rubber films obtained from separated latexes from different *Hevea brasiliensis* clones: (**a**) IAN873; (**b**) Dongfang93114; (**c**) Reyan73397.

**Figure 8 ijms-23-08880-f008:**
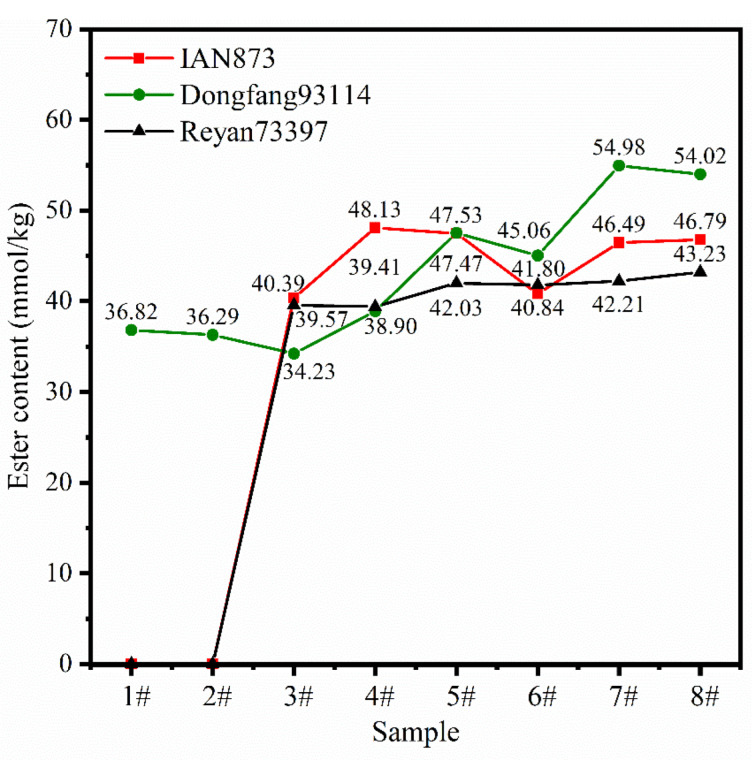
Ester content of rubber obtained from separated latexes of different *Hevea brasiliensis* clones.

**Figure 9 ijms-23-08880-f009:**
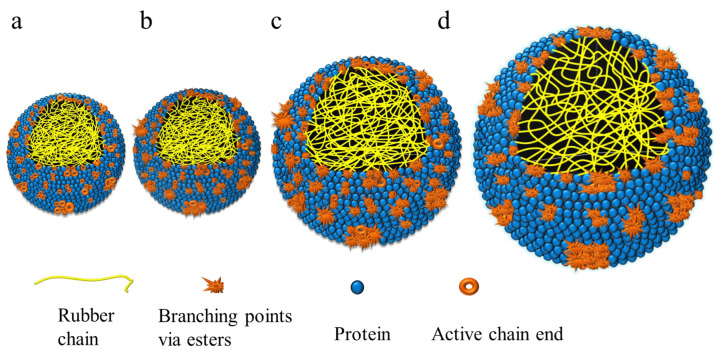
Presumed structure of representative rubber particles: (**a**) SRPs in IAN873 and Reyan73397; (**b**) SRPs in Dongfang93114; (**c**) rubber particles in sample 3#; (**d**) rubber particles with the most branching points.

**Figure 10 ijms-23-08880-f010:**
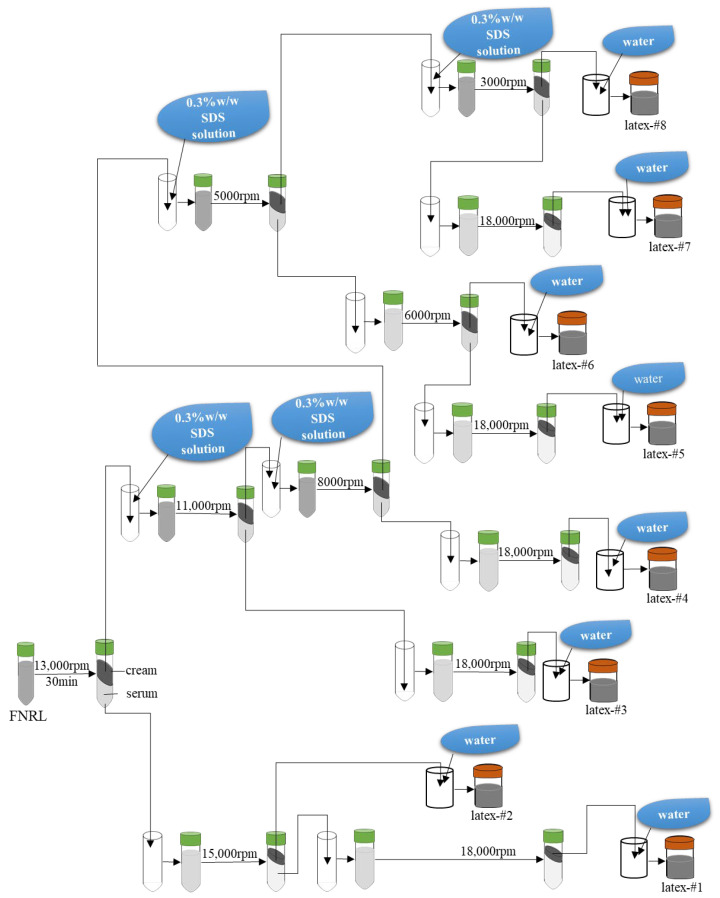
Fractionation process of FNRL by centrifugation at different revolving speed rates.

**Table 1 ijms-23-08880-t001:** Average particle size of separated latexes from different *Hevea brasiliensis* clones.

Sample	IAN873		Dongfang93114		Reyan73397	
	Average Particle Size (nm)	Content (%)	Average Particle Size (nm)	Content (%)	Average Particle Size (nm)	Content (%)
Latex-#1	137	1.8%	138	7.8%	130	6.8%
Latex-#2	174	11.4%	196	17.2%	194	9.1%
Latex-#3	223	4.0%	250	5.8%	234	2.1%
Latex-#4	366	7.0%	381	2.1%	326	6.1%
Latex-#5	619	2.4%	611	11.1%	552	2.6%
Latex-#6	841	33.1%	857	15.4%	801	34.1%
Latex-#7	971	32.7%	929	38.7%	995	29.4%
Latex-#8	1181	7.6%	1236	1.9%	1255	9.8%

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
