# Peer review of "Influence of Clones on Relationship between Natural Rubber and Size of Rubber Particles in Latex"

_ijms, 2022, doi:10.3390/ijms23168880_

Round 1

Reviewer 1 Report

What is the contribution to research the paper you are presenting?

What kind of improvement or applicability does your research present? Include this information in the conclusions

What does the presence of nitrogen contribute to the present materials analyzed, why the presence of Nitrogen is rellevant for the present paper?

How were the clones created? Why they were created, for what purpose? This information should be summarized in the abstract

Author Response

Comments to the Author: What is the contribution to research the paper you are presenting?

Authors’ response: We are very grateful for your question. The study reveals the influence of clone on the relationship between characteristics of natural rubber and the size of rubber particles in natural rubber latex. And then characteristics of natural rubber such as nitrogen content, ester content and so on can be adjusted by selecting clone and particle size so that meet applications in different fields.

Comments to the Author: What kind of improvement or applicability does your research present? Include this information in the conclusions

Authors’ response: Thank you very much for your comments. This research provides a method to adjust the structure and non-rubber component content by selecting clone and particle size. The corresponding content has been added to the conclusions in the revised manuscript.
Comments to the Author: What does the presence of nitrogen contribute to the present materials analyzed, why the presence of Nitrogen is relevant for the present paper?

Authors’ response: Thanks a lot for your question. The nitrogen content reflects the protein content and protein is one of the major non-rubber components in natural rubber. It plays an indispensable role in excellent characteristics of natural rubber. Rubber particle is assembly formed by a core-shell structure, and the mixed protein−lipid layer forms the shell. In addition, previous works have reported that protein can promote vulcanization and ageing resistance of natural rubber.
Comments to the Author: How were the clones created? Why they were created, for what purpose? This information should be summarized in the abstract

Authors’ response: We are full of gratitude for your advice. The clones are created through vegetative propagation to shorten breeding cycle and improve production efficiency (Natural rubber grows naturally with seed reproduction and cross-pollination, so the breeding cycle is long). IAN873, Dongfang93114 and Reyan73397 are the major clones planted in China for their high yield and excellent cold resistance. The information is summarized in the abstract in revised manuscript.

Reviewer 2 Report

Dear Authors

1) specify the aim

2) add whether highlights or graphical abstract or both

3) add more international references outside Asia

4) improve stylistics and grammar; Figures visual outline

Author Response

Comments to the Author 1): specify the aim

Authors’ response: We are very grateful for your advice. And the aim for this study is to reveal the influence of clone on the relationship between characteristics of natural rubber and the size of rubber particles in natural rubber latex. And then characteristics of natural rubber such as nitrogen content, ester content and so on can be adjusted by selecting clone and particle size so that meet applications in different fields.

Comments to the Author 2): add whether highlights or graphical abstract or both

Authors’ response: Thank you very much for your question. Highlights and graphical abstract are both added.

Comments to the Author 3): add more international references outside Asia

Authors’ response: Thanks a lot for your suggestion. More references from France, Spain, Poland and Romania have been added and included and we would also appreciate your further specific suggestions.
Comments to the Author 4): improve stylistics and grammar; Figures visual outline

Authors’ response: Thank you very much for your suggestion. The corresponding content have been improved and modified. We will be also grateful for your further comments.

Round 2

Reviewer 1 Report

Please add the new modified manuscript version with the highlighted changed in other colour, so I will can check all the changes that you have performed in the modified version.

The aspects commented and asked to authors must be included as changes and improvements in the text article reviewed, so is necessary not only the reviewer answer, is also necessary to include the improvements about each point in the paper

Author Response

Thank you very much for your comments.  We have accordingly revised the manuscript and the modified content are highlighted with yellow shading.

 point-by-point response:

Comments to the Author: What is the contribution to research the paper you are presenting?

Authors’ response: We are very grateful for your question. The study reveals the influence of clone on the relationship between characteristics of natural rubber and the size of rubber particles in natural rubber latex. And then characteristics of natural rubber such as nitrogen content, ester content and so on can be adjusted by selecting clone and particle size so that meet applications in different fields. The corresponding contents are highlighted with yellow shading in Introduction and Abstract.

Comments to the Author: What kind of improvement or applicability does your research present? Include this information in the conclusions

Authors’ response: Thank you very much for your comments. This research provides a method to adjust the structure and non-rubber component content by selecting clone and particle size. The corresponding content has been added to the conclusions in the revised manuscript and is highlighted with yellow shading.
Comments to the Author: What does the presence of nitrogen contribute to the present materials analyzed, why the presence of Nitrogen is relevant for the present paper?

Authors’ response: Thanks a lot for your question. The nitrogen content reflects the protein content and protein is one of the major non-rubber components in natural rubber. It plays an indispensable role in excellent characteristics of natural rubber. Rubber particle is assembly formed by a core-shell structure, and the mixed protein−lipid layer forms the shell. In addition, previous works have reported that protein can promote vulcanization and ageing resistance of natural rubber. The corresponding content is highlighted with yellow shading in Introduction.
Comments to the Author: How were the clones created? Why they were created, for what purpose? This information should be summarized in the abstract

Authors’ response: We are full of gratitude for your advice. The clones are created through vegetative propagation to shorten breeding cycle and improve production efficiency (Natural rubber grows naturally with seed reproduction and cross-pollination, so the breeding cycle is long). IAN873, Dongfang93114 and Reyan73397 are the major clones planted in China for their high yield and excellent cold resistance. The information is summarized in the Abstract in revised manuscript and the corresponding content is highlighted with yellow shading.

Reviewer 2 Report

now it is ready after minor check outs to be published

Author Response

Thank you very much for your attention and comments. The Manuscript has been modified and checked.

Round 3

Reviewer 1 Report

Conclusions must include mores aspects and details about the What kind of improvement or applicability does your research present? and which contributions presents the present research?

Please highlight the changes in the resubmiteed manuscript

Author Response

Comments to the Author: Conclusions must include mores aspects and details about the What kind of improvement or applicability does your research present? and which contributions presents the present research?

Authors’ response: We are full of gratitude for your comments.

It reveals a method to adjust the structure (e.g., molecular weight, molecular weight distribution and branching structure) and non-rubber component content (e.g., nitrogen content and ester content) by selecting clone and particle size. And then provide new perspectives to adjust the properties of natural rubber (e.g., cure characteristics, mechanical properties and ageing-resistant performance) so as to promote the application of natural rubber in different conditions by artificial selection of natural factors (clone and particle size).

The corresponding content has been included in the latest revised manscript and is highlighted with yellow shading. We will also be grateful for your further and specific comments as well as suggestions.